# Overfeeding and Substrate Availability, But Not Age or BMI, Alter Human Satellite Cell Function

**DOI:** 10.3390/nu12082215

**Published:** 2020-07-24

**Authors:** Dane W. Fausnacht, Ryan P. McMillan, Nabil E. Boutagy, Ryan A. Lupi, Mordecai M. Harvey, Brenda M. Davy, Kevin P. Davy, Robert P. Rhoads, Matthew W. Hulver

**Affiliations:** 1Department of Human Nutrition, Foods, and Exercise, Virginia Polytechnic Institute and State University, Blacksburg, VA 24061, USA; mcmillr@vt.edu (R.P.M.); lryan91a@vt.edu (R.A.L.); mordecai.m.harvey@vanderbilt.edu (M.M.H.); bdavy@vt.edu (B.M.D.); kdavy@vt.edu (K.P.D.); hulvermw@vt.edu (M.W.H.); 2Virginia Tech Metabolic Phenotyping Core, Virginia Polytechnic Institute and State University, Blacksburg, VA 24061, USA; 3Department of Animal and Poultry Sciences, Virginia Polytechnic Institute and State University, Blacksburg, VA 24061, USA; rhoadsr@vt.edu; 4Department of Pharmacology, Yale University School of Medicine, Yale University, New Haven, CT 24061, USA; nabil.boutagy@yale.edu; 5Vascular Biology and Therapeutics Program, Yale University School of Medicine, Yale University, New Haven, CT 24061, USA

**Keywords:** overfeeding, aging, skeletal muscle

## Abstract

Satellite cells (SC) aid skeletal muscle growth and regeneration. SC-mediated skeletal muscle repair can both be influenced by and exacerbate several diseases linked to a fatty diet, obesity, and aging. The purpose of this study was to evaluate the effects of different lifestyle factors on SC function, including body mass index (BMI), age, and high-fat overfeeding. For this study, SCs were isolated from the vastus lateralis of sedentary young (18–30 years) and sedentary older (60–80 years) men with varying BMIs (18–32 kg/m^2^), as well as young sedentary men before and after four weeks of overfeeding (OVF) (55% fat/ + 1000 kcal, *n* = 4). The isolated SCs were then treated in vitro with a control (5 mM glucose, 10% fetal bovine serum (FBS)) or a high substrate growth media (HSM) (10% FBS, 25 mM glucose, and 400 μM 2:1 oleate–palmitate). Cells were assessed on their ability to proliferate, differentiate, and fuel substrate oxidation after differentiation. The effect of HSM was measured as the percentage difference between SCs exposed to HSM compared to control media. In vitro SC function was not affected by donor age. OVF reduced SC proliferation rates (–19% *p* < 0.05) but did not influence differentiation. Cellular proliferation in response to HSM was correlated to the donor’s body mass index (BMI) (*r*^2^ = 0.6121, *p* < 0.01). When exposed to HSM, SCs from normal weight (BMI 18–25 kg/m^2^) participants exhibited reduced proliferation and fusion rates with increased fatty-acid oxidation (*p* < 0.05), while SCs from participants with higher BMIs (BMI 25–32 kg/m^2^) demonstrated enhanced proliferation in HSM. HSM reduced proliferation and fusion (*p* < 0.05) in SCs isolated from subjects before OVF, whereas HSM exposure accelerated proliferation and fusion in SCs collected following OVF. These results indicated that diet has a greater influence on SC function than age and BMI. Though age and BMI do not influence in vitro SC function when grown in controlled conditions, both factors influenced the response of SCs to substrate challenges, indicating age and BMI may mediate responses to diet.

## 1. Introduction

Muscle mass correlates with increased metabolic rate and lower rates of obesity, as well an improved overall quality of life [1,2,3,4]. Increased muscle mass and strength contributes more to overall health in elderly populations, as it increases mobility and decreases risk of injury from falls [4,5]. However, aging is associated with a 1% decrease in lean mass and 3–4% muscular strength per year [6,7]. In extreme cases of sarcopenia, the accelerated muscle loss associated with aging and frailty syndrome, there is documented dysfunction in muscle progenitor cells, which inhibit regrowth [5,8]. Loss of muscle mass is also a common consequence of other age-related chronic diseases including, obesity, cancer, diabetes, chronic obstructive pulmonary disease and heart disease [9,10].

To explain the loss in muscle mass associated with age and disease, many researchers have assessed the mechanisms of tissue repair, specifically those mediated by the satellite cell (SC) [11,12,13]. SCs are muscle progenitor cells responsible for replacing atrophied or damaged fibers [14]. These cells replicate and form new fibers or fuse to existing fibers, donating undamaged DNA and increase transcription and protein translation. This is done through a myriad of transcription factors [14,15]. These transcription factors must be expressed in a timed and coordinated effort to allow for proper proliferation, differentiation, and fusion of SCs.

SCs commonly exist as quiescent cells located between the sarcolemma and basement membrane of skeletal muscle [16] and can be activated under stress, such as contraction-mediated muscle damage [17] and metabolic dysfunction [11]. The involvement of SCs in muscle function holds importance due to their continual integration into differentiated tissue [18], and because of this, many interventions have targeted improving SC incorporation into muscle [19,20]. SCs are influenced by a variety of inflammatory factors [21,22,23], hormonal variations [24,25], and epigenetic modifications [26], all of which must be properly coordinated during the process of myogenesis and muscle tissue maintenance.

SC function can be altered by disease and negative lifestyle factors, such as a sedentary lifestyle [27] and obesity [28]. The complex and dynamic life cycle of the SC creates many possible points for mechanistic failure: chiefly, maintaining quiescence and proper SC pool size, proliferating to adequate number upon activation, or properly differentiating and fusing to form and repair healthy fibers. SCs undergo a physical change in structural phenotype, from stem cell to tissue, but, also, increases in oxidative capacity and changes in substrate preferences [29,30,31].

Diet-induced obesity (DIO) in rodents is associated with altered SC function, specifically, blunted activation and proliferation and diminished insulin receptor signaling [28,32,33]. Some of the phenotypic changes associated with DIO, such as insulin resistance and altered cytokine signaling, are known to be retained in cultured SCs [33,34]. The ability to separate the effects of the high-fat, high-calorie diets and the physiological state of obesity on SC function is challenging. The effect of diet alone on SC function has lacked extensive research and little information is available in humans. High-fat and hypercaloric diets induce an inflammatory state [35], even in otherwise healthy individuals [35,36], which may reduce the myogenicity of quiescent and activated SCs [11,28].

This study sought to determine the influence of donor diet, age, and body mass index (BMI) on SC function as defined by their ability to proliferate, and fuse into multinucleated fibers. SCs were collected from: young sedentary lean, young sedentary overweight/obese, aged sedentary lean, and high-fat/hypercaloric fed young sedentary lean donors. 

## 2. Materials and Methods

Human research participants: Skeletal muscle biopsies were obtained under Virginia Polytechnic and State University, Internal Review Board-approved protocols (#12–652 [37] and #11–077 [38]), and SCs were subsequently isolated (described below). All SC donors provided written consent. Body composition (body fat, (BF)) was measured using dual energy X-ray absorptiometry (DEXA). Blood metabolite measures were obtained after a minimum eight-hour fast immediately prior to muscle biopsies as previously described in Osterberg et al. [38] and Flack et al. [37]. Body mass index was between 18–32 kg/m^2^, and body fat was between 11–33%. All participants were male and sedentary (≤2 days/week, ≤20 min/day of low-intensity physical activity). Participants were grouped as sedentary normal weight (SedLn, 18–35 years, BMI 18–25 kg/m^2^, BF <25%, *n* = 18), sedentary overweight or obese (SedOvOb, 18–35 years, BMI >25 kg/m^2^, BF >25%, *n* = 5), or sedentary aged (SedOld, 60–80 years, BMI 18–35 kg/m^2^, *n* = 20). The SedOld group had an average appendicular skeletal muscle index (ASMI, appendicular lean mass/m^2^) of 9.70 ± 0.13 kg/m^2^ (previously reported by Flack et al. [37]), indicating they were not yet sarcopenic (ASMI <7.0 kg/m^2^) [39,40].

Dietary intervention: Using a controlled feeding approach, participants were provided with a 10–14 day lead-in diet immediately followed by a high-fat hypercaloric diet (OVF) for four weeks as described by Osterberg et al. [38]. The lead-in diet was isocaloric to habitual dietary intake and consisted of 30% fat (8% saturated), 50% carbohydrate, and 20% protein. The OVF delivered 1000 kcal above what was provided during the lead-in period with a macronutrient composition of 55% fat (25% saturated), 30% carbohydrate, and 15% protein. Participants did not experience significant weight gain or changes in fasting blood glucose or lipid profiles [38]. SCs were isolated (*n* = 4) from skeletal muscle biopsies collected following the lead-in diet (before OVF) and following OVF diet and used for the studies described herein.

### 2.1. Cell Culture

SCs were collected from tissue biopsies of the vastus lateralis muscle of participants as described in Muoio et al. [41]. SCs were cultured and differentiated into multinucleated myotubes as previously described [42]. Cells were seeded and allowed to proliferate in a growth medium (DMEM with 5 mM glucose, 5% fetal bovine serum, rhEGF (10 n/mL), and dexamethasone (0.4 ug/mL), SkGM^TM^, Lonza, Rockland, ME, USA) for 5 days until they reached 60–80% confluency. After proliferation, cells were differentiated for seven days in a low-serum media (DMEM with 5 mM glucose and 2% horse serum) with differentiation media replaced every 2 days. Amphotericin-B (0.5 ug/mL), antimitotic (for penicillin 100 U/mL and streptomycin 100 ug/mL), and gentamycin (0.01 mg/mL) were added for protection against bacterial and fungal contamination.

### 2.2. High Substrate Media Challenge

High substrate media (HSM) was applied during the proliferation stage of SCs. HSM contained the same ingredients as the control growth media with additional supplementation of 400 mM, 2:1, oleate–palmitate complexed with 5% BSA (bovine serum albumin), and the glucose concentration increased from 5 mM to 25 mM (typical fasting blood glucose of T2DB patients is a ~7.0 mM) [43]. GLUT1, being the primary glucose transporter of proliferating SCs [44], reaches maximal transport capacity at concentrations above ~6 mM [45]. While higher than in vivo measures, glucose concentrations of 25 mM should ensure SCs experience uptake similar to cells exposed to in vivo hyperglycemia without the need for frequent media changes). Control growth media and HSM were replaced simultaneously as shown in Table 1. For the proliferation assay, HSM was applied one day after the cells were plated and replaced after 4 days. For all differentiation protocols, HSM was applied one day after the cells were plated, and cells were proliferated in HSM until they reached ~60–80% confluency (5 days) at which point both the cells exposed to control media and HSM were differentiated identically in a low-serum differentiation media. All high-substrate-challenged cells and matching control cells were grown on the same plate and differentiated simultaneously. SedLn SC were used as the control for HSM experiments.

### 2.3. Proliferation Assay

Cells were counted with a hemocytometer and each cell line was seeded in triplicate at 2500 cell/well in 96-well plates. Cells used for proliferation measures were generated after their third passage (P3). Plates were collected on days 1 through 6, and nuclear material was counted using the CyQUANT^®^ NF Cell Proliferation Assay Kit (Thermo Fisher Scientific, Rockford, IL, USA, Cat # C35007). Doubling times were calculated using the PRISM exponential growth equation software [46] (GraphPad Software Inc., San Diego, CA, USA). Proliferation rate is reported as doublings per day (dbs./day ± SEM).

### 2.4. Myogenic Index

Myogenic index methods were adapted from Kamli et al. [47]. Cells were seeded at 25,000 cell/well in 12-well plates and allowed to proliferated for 4 days prior to differentiation (this seeding volume yields approximately 30–50% confluency after 24 h). Cells used for differentiation measures were generated after 4 passages (P4). Cells were fixed in 4% formaldehyde for 15 min at room temperature and stained red with eosin, while nuclei were stained green with CyQUANT Dye (Thermo Fisher Scientific, Rockford, IL, USA, Cat # C35007). Cells were imaged at 10× magnification. Each cell line was seeded in three wells per plate with one image taken per well; values were taken from the average of the three images. Myogenic index was calculated as the percentage of nuclei within multinucleated fibers compared to total observed nuclei, as described by Kamli et al. [47]. Myogenic index is reported as both simple fusion (proportion of nuclei in fibers with 2 or more nuclei per fiber divided by total nuclei) and robust fusion (proportion of nuclei in fibers with 10 or more nuclei per fiber divided by total nuclei). With simple nucleation being an indication that the fusion process has begun and robust nucleation indicating that more complete myotubes have formed. Myogenic index scores for simple and robust nucleation are reported as a percent ± SEM.

### 2.5. Substrate Oxidation

To determine the effects of the diet and substrate treatments on SC metabolism, radio labeled substrate metabolism was performed on differentiated cells as previously described in Frisard et al. [48]. Cells were seeded at 20,000 cells/well in 12-well plates and proliferated for 6 days in control or HSM media before 7 days of differentiation. Complete fatty-acid oxidation (FAO) (fully oxidized palmitate to CO2), incomplete FAO (acid soluble metabolites), total FAO (complete + incomplete), and the ratio of complete FAO to incomplete FAO, as well as glucose oxidation, were measured in cells after seven days of differentiation.

### 2.6. RNA Extraction and qRT-PCR

Cells were grown in either control or HSM for 4 days prior to differentiation as stated. RNA was collected on days 0, 1, 2, and 7 of differentiation. RNA was extracted using a RNeasy Mini Kit (Qiagen) and DNase I treatment (Qiagen, Valencia, CA, USA), according to the manufacturer’s instructions. qRT-PCR was performed using a ViiA 7 Real-Time PCR System (Thermo Fisher Scientific, Waltham, MA, USA) and TaqMan Universal PCR Master Mix, used according to the manufacturer’s specifications (Applied Biosystems™, Foster City, CA, USA). Target gene expression in human skeletal muscle primary-cell culture was normalized to peptidyl-prolyl cis-trans isomerase B (PPIB) RNA levels. TaqMan probes were purchased as pre-validated assays for Pax7(Hs00242962_m1), MyoD (Hs00159528_m1), Myogenin (Hs01072232_m1), and PPIB (Hs00168719_m1) (Applied Biosystems™, Foster City, CA, USA). Relative quantification of target genes was calculated using the ΔCT method. Derivation of the ΔCT equation has been described in Applied Biosystems User Bulletin no. 2 (P/N 4303859).

### 2.7. Statistical Analysis

One-way ANOVA was used to compare proliferation rates between the SedLn vs. SedOvOb and SedLn vs. SedOld with a Tukey post hoc test for multiple comparisons. SCs isolated before vs. following OVF and control vs. HSM samples were compared using a paired *t*-test. Differentiation and the myogenic index, as well as RNA expression, were analyzed using repeated-measures ANOVA and a Tukey’s post hoc analysis to compare daily differences. Pearson’s correlations were used to explore relationships between SC function and age, bodyfat percentage, and BMI. The HSM effect is reported as percent change from control values. Data is presented as mean ± SEM and statistical significance was set at a *p*-value of ≤0.05. The samples were taken during two different Virginia Polytechnic Institute and State University, Internal Review Board-approved studies (#12–652 and #11–077). SC-culture approval was further granted under #14–1234. Written consent was obtained for both studies.

## 3. Results

Influence of bodyweight/adiposity on SC function: SC cultures were obtained from 23 young (23.6 ± 0.7 years) participants with BMI ranging from 20–33 kg/m^2^ (average (avg) 24.2 ± 0.6 kg/m^2^, avg bodyfat 21.3 ± 1.5%) (grouped anthropometric and blood measures can be found in Table 1). There were no correlations between SC donor BMI and measures of SC proliferation rate (Figure 1B) or fusion index scores. When grouped by lean (SedLn, BMI 18–25 kg/m^2^, avg body fat 18.6 ± 1.4%, *n* = 18) vs. overweight/obese (SedOvOb, BMI >25 kg/m^2^, avg bodyfat 29.6 ± 2.8%, *n* = 5), there were no differences in proliferation (*p* = 0.59) (Figure 1A). SedLn did exhibit higher simple fusion overall (repeated-measures ANOVA group effect of *p* = 0.04), which was primarily driven by reduced spontaneous fusion (fusion prior to the addition of differentiation media) in the SedOvOb group (Figure 1C). There was no effect of BMI on robust fusion between these groups (Figure 1D). A negative correlation was observed between SC proliferation and adiposity, with higher donor bodyfat percentages having lower proliferation rates (Figure 1E).

Influence of age on SC function: SC cultures of 38 subjects were used in this analysis of SedLn vs. SedOld SC function. Of those, 18 were young, and 20 were older. The SedOld group had significantly higher BMIs (26.5 ± 0.6 kg/m^2^, *p* < 0.01) and body fat levels (28.1 ± 1.3%, *p* < 0.01) compared to the younger SedLn group, but both groups participated in minimal physical activity (≤2 days/week, ≤20 min/day of low-intensity physical activity) (comparison of SedLn and SedOld participants can be found in Table 1). There was no significant difference in proliferation rate between the young and aged groups (*p* = 0.73, Figure 2A). In sedentary subjects, age did not correlate with proliferation rate (*p* = 0.23) (Figure 2B). Within the SedOld group, there was no correlation between proliferation rate and BMI (*p* = 0.39, Figure 2A). 

The SedLn and SedOld groups achieved similar simple fusion (2+ nuclei) rates throughout differentiation (Figure 2C). Both groups experienced a significant amount of spontaneous simple fusion prior to differentiation, with both groups achieving greater than 45% fusion prior to the addition of differentiation media. However, robust fusion did not occur in either group until the addition of differentiation media at which point the SedOld group achieved higher overall rates of robust fusion (*p* = 0.03, Figure 2D). Neither BMI nor bodyfat percentage had any correlation with myogenicity within the SedOld group. Metabolically, complete and total FAO (Figure 3A,C) as well as glucose oxidation (Figure 3E) were not different between differentiated SedLn and SedOld SCs. SedOld cells did exhibit reduced incomplete FAO (acid soluble metabolites) (Figure 3B) (*p* = 0.05), and, thus, had higher rates of oxidative efficiency (*p* = 0.03) (Figure 3D). 

Effects of overfeeding on SC function: As a result of the OVF intervention, participants experienced significant weight gain of 1.8 ± 0.5 kg (*p* = 0.03). There was no change in fasting blood glucose (4.8 ± 0.1 nM pre vs. 4.6 ± 0.7 nM post, *p* = 0.70) or triglycerides (115.0 ± 16.2 mg/dL pre vs. 113.0 ± 7.4 mg/dL post, *p* = 0.9). HDL (45.7 ± 5.2 mg/dL pre vs. 58.3 ± 4.5 mg/dL post, *p* = 0.06), LDL (91.7 ± 16.4 mg/dL pre vs. 109.3 ± 22.3 mg/dL post, *p* = 0.07), and total cholesterol (147.0 ± 13.0 mg/dL pre vs. 172.3 ± 21.6 mg/dL post, *p* = 0.08) trended on changes but were not statistically significant. SCs collected after 4 weeks of OVF [38] exhibited a 20.9% reduction in their proliferation rate compared to SCs from the same donor collected before OVF (*p* = 0.03) (Figure 4A). However, OVF had no effect on SC fusion rates prior to or throughout differentiation (Figure 4B,C).

Effects of HSM on SedLn SC Function: SedLn (BMI 18–25) SCs treated with HSM demonstrated reduced proliferation (4.83 ± 1.95%) compared to control conditions (*p* < 0.01) (Figure 5A). HSM treatment also reduced fusion index scores in SedLn SCs (*p* < 0.05) (Figure 5A,B). HSM suppressed the spontaneous fusion (cellular fusion which occurs prior to the application of differentiation media) of simple multinucleated fibers by 21.7% (*p* = 0.03). The effects of HSM exposure during the proliferative phase suppressed fusion after eight days of differentiation reducing simple and robust fiber nucleation by 15.7% (*p* = 0.02) and 16.5% (*p* = 0.03), respectively. Though there is a reported reduction in both proliferation and fusion prior to differentiation treatment it is possible that cells which attempted to spontaneously differentiate perished. Due to the proliferation assay only counting live cells, those that did not survive spontaneous differentiation result in a reduced measured proliferation.

Capacity for fatty-acid oxidation improved in differentiated SCs exposed to HSM during their proliferative phase (Figure 6A,C,D). Complete palmitate oxidation to CO_2_, total FAO, and the ratio of complete to incomplete FAO (CO2:ASM) all improved (*p* < 0.05) (Figure 6A,C,D). HSM exposure had no effect on glucose oxidation (Figure 6E). A metabolic shift towards FAO is typical in differentiated SCs [49,50], however, in our model, cells exposed to HSM exhibited reduced rates of fusion in concert with increased FAO (Figure 5E,F). This indicates a disconnect between changes in fatty-acid oxidation and differentiation, highlighting that SCs are metabolically adapting to available substrate but experiencing a deficit in the progression to differentiation. 

Pax-7, MyoD, and MyoG RNA-expression varied throughout proliferation and differentiation (Figure 7A–D). At day one of differentiation, SCs exposed to HSM exhibited a more rapid cessation (15% vs. 48%, *p* < 0.05) of Pax7 expression (*p* < 0.01) and increased (62%, *p* < 0.05) expression of myogenin (*p* < 0.01), but with no change in MyoD compared to control SCs. After day one, HSM exposure had no effect on Pax7, MyoD, or myogenin expression.

Effect of HSM on Obese, Older, and OVF SCs: Response to HSM was influenced by BMI, age, and the OVF dietary intervention. While SCs isolated from the SedLn group experienced reduction in proliferation due to HSM exposure, those SCs isolated from the SedOvOb and SedOld group exhibited increases in the proliferation rate when exposed to HSM (6.68 ± 1.23% and 6.56 ± 1.01%, respectively) (*p* < 0.05). In pre-OVF SCs, HSM treatment reduced proliferation by −5.35 ± 1.14%. Conversely, for SCs collected from the same individuals post-OVF, HSM exposure caused an increased proliferation rate of 7.17 ± 3.65% (*p* < 0.05) (Figure 5B). The same response was seen in fusion index scores, where HSM diminished fusion in SCs collected pre-OVF (*p* < 0.05), yet HSM treatment increased simple and robust fusion of SCs collected post-OVF (Figure 4B,C).

## 4. Discussion

The primary findings of these studies are: (1) a high fat/hypercaloric feeding intervention in humans reduced cultured SC proliferation; (2) in sedentary participants age did influence in vitro SC function under control conditions, and (3) elevated concentration of fatty acids and glucose during SC proliferation have differential effects on SC function that may be influenced by the BMI, age, and diet of the SC donor. The effects of overfeeding on SCs may be linked to substrate availability as in vivo OVF and in vitro HSM treatments, resulting in similar changes to SC function.

SCs may adapt to the substrate levels of their environment as demonstrated by the opposing responses to the HSM treatment of lean vs. overweight/obese SCs and SCs collected before and following overfeeding. Both BMI and bodyfat percentage showed some influence of SC function with overweight/obese SCs presenting with reduced fusion and bodyfat, negatively correlating with proliferation. These data, when taken with the OVF and HSM results from this study and previously demonstrated effects of DIO models [3,32], show that substrate levels have a potent effect on SC function. Response of SCs to differing substrate levels does appear adaptable. The SedOvOb and SedOld groups had higher fasting blood metabolic measures compared to the SedLn group, and both groups appeared to respond positively to the HSM treatment compared to the SedLn group, whose SC function was blunted by HSM.

Donor age appeared to have no influence on in vitro function of SCs in humans after physical maturity. Donor age caused no differences in an SCs ability to proliferate, fuse, or differentiate into metabolically functional tissue. These results are contrary to findings from previous human and rodent models that suggest aging alters the function of SCs at their core, as aging has been shown to disrupt SC function in vivo [12,50,51,52]. However, the in vitro data in this current study suggest that these disruptions must be more attributed to environmental factors, for when the young and aged cells are grown in equal conditions, the dysfunction disappears. The idea that the environment is the main cause of SC dysfunction is evidenced by previous studies, which have used young sera to restore function in aged SCs [53]. Some studies have even shown that improvements in the aged environment through exercise can restore SC function [54].

Previous studies have demonstrated that in vivo phenotypes of metabolic disease, such as insulin resistance [33,34,55] and oxidative stress [56], can be retained in cultured SCs. In this study, four weeks of overfeeding significantly reduced SC proliferation. While the overfeeding diet from this study did not significantly alter SC differentiation (as measured by cellular fusion), the reduction in proliferative potential was similar to deficits observed in disease models of muscular dystrophy, such as mdx mice SCs [57,58]. This similarity highlights the degree to which diet can affect SC function. However, it should be noted that the effects of the OVF diet on SC function are preclinical, as participants did not undergo any loss of muscle mass or function due to the diet.

SC myogenesis is transcriptionally regulated through the sequential expression of key transcription factors (TFs): Pax-7 [18] (quiescence/activation), MyoD [15] (proliferation/differentiation), and myogenin [15] (fusion/contractile protein expression). However, many more TFs play a role, and all are susceptible to manipulation by disease [14] and lifestyle factors [15,23]. The myogenic TFs MyoD [49] and myogenin [59], aside from inducing fusion and protein synthesis, also facilitate an oxidative phenotype by promoting the expression of genes involved in fatty acid uptake and beta-oxidation [49,59]. The HSM media in this study altered expression patterns of both Pax-7 and myogenin. This was not unexpected as previous studies have used lipid enriched media to mechanistically induce SC differentiation through the expression of MyoD and associated regulation of SC fat oxidation [60]. That the expression of these TFs was modulated by substrate supply suggests a potential mechanistic link between substrate availability and SC function [50,60]. The opposing responses of lean and overweight/obese SC may be due to both substrate availability and lifestyle factors having regulatory control over myogenic TF expression. The differential relationship between nutrient intake and myogenic TF expressed in lean vs. obese populations should be further examined.

Overfeeding and excessive substrate significantly diminished SC function as seen in previous rodent models [28,32,34]. The evidence that SC function is adaptable to environmental substrate levels hints at a possibility that the negative effects of DIO on SC function may be restored with proper dietary interventions. Evidence suggests poor diet and increased adiposity may mechanistically act through disruptions in mitochondrial function and lipid metabolism [33,50,60,61]. High-carbohydrate diets also reduce mitochondrial function and lipid metabolism [62,63] and, therefore, it is expected that they would reduce SC function as well, however, this effect is typically less potent compared to high-fat diets [62,63,64].

The effects of obesity on SC function may exist more dominantly in vivo, as, once the cells are removed from the obese environment, they are no longer subjected to disrupted adipokine signaling [65]. Though a small subset of overweight/obese SCs were used, donor BMI showed no relationship with in vitro measures of SC proliferation. There was, however, a significant negative relationship between donor bodyfat percentage and SC proliferation rate; as bodyfat is increased, adipose tissue uptake capacity is maximized and lipid is disproportionally stored in muscle, increasing inflammation and tissue function [66,67]. 

The general increase in confounding conditions that coincide with aging populations (i.e., obesity, diabetes, cardio vascular disease metabolic syndrome, etc.) make it difficult for any human study to isolate the true effects of chronological aging. Large differences in adiposity between young and aged counterparts is a common complication seen in aging studies, as it is difficult to recruit a lean aged population. Even in the subject demographics for this work, there was a significant difference between the adiposity of the young and aged group. With the high prevalence of body fat differences in aging studies, we must ask if the differences currently attributed to age are truly due to the chronological age of the participants or could the variation be more accurately ascribed to the adiposity of the aged population.

Perhaps one of the most intensive studies designed to elicit the mechanistic difference between young and old SCs is that of the work done by Roberts et al., published in 2011. This work found that Cyclin D1 and MyoD were determined to have different expressional patterns in young vs. aged SCs. In response to exercise, MyoD increased in the young group as expected, but there was no increase in the aged group. [13] While this finding was attributed to the effects of aging, it is important to note that the bodyfat levels between the young and old group were significantly different with the 10 young subjects at 15.4% (+/−2.9%) and the 10 older subjects at 27.4% (+/−1.9%). These numbers place the young subjects in what is considered a “healthy” range for bodyfat, while the aged group would likely be considered “unhealthy” in the overweight/obese ranges. This makes it difficult to attribute the SC expressional differences to age as opposed to differences in body composition. The bodyfat content discrepancy is a common occurrence, as it is extremely difficult to recruit two groups of human participants where the only difference is the subjects’ age.

## 5. Conclusions

These results highlight the resiliency and adaptability of SCs, as age did not influence SC function. SCs were negatively affected by BMI, bodyfat, and overfeeding with reduced proliferation and fusion. While overfeeding causes a variety of systemic effects, such as inflammation and altered hormonal signaling, the results of the high substrate in vitro treatment indicate that changes to substrate levels alone are enough to alter SC function. This data suggests a reasonable viability for the use of dietary interventions to alter SC function. Age and BMI were influential in the SCs’ response to the substrate challenge, with SCs from both older and heavier donors preferring higher substrate. This is an indication that older and more obese populations may respond differently to dietary interventions designed to influence muscle function.

## Figures and Tables

**Figure 1 nutrients-12-02215-f001:**
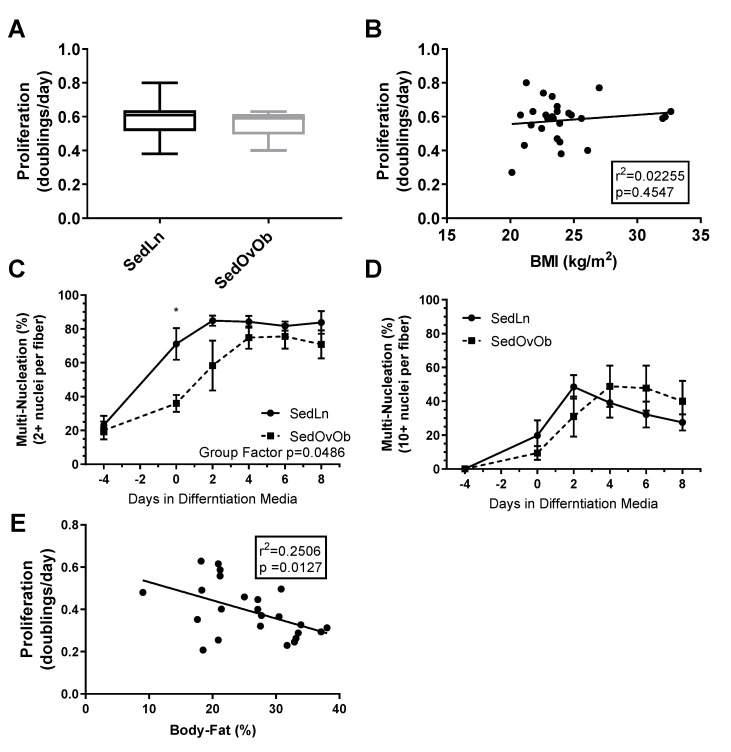
Proliferation rates (doubling/day) of satellite cells (SCs) isolated from sedentary lean (SedLn) (*n* = 18, BMI 18–25 kg/m^2^) and sedentary overweight/obese (SedOvOb) (*n* = 5, BMI >25 kg/m^2^) participants (**A**). Correlation of SC proliferation rate and donor BMI (**B**). Fusion index scores ((**C**), 2+ nuclei per fiber, (**D**), 10+ nuclei per fiber) of SC from SedLn and SedOvOb donors. (**E**). Correlation of proliferation rate and donor bodyfat percentage (*, ** indicating a *p*-value of <0.05, <0.01, respectively).

**Figure 2 nutrients-12-02215-f002:**
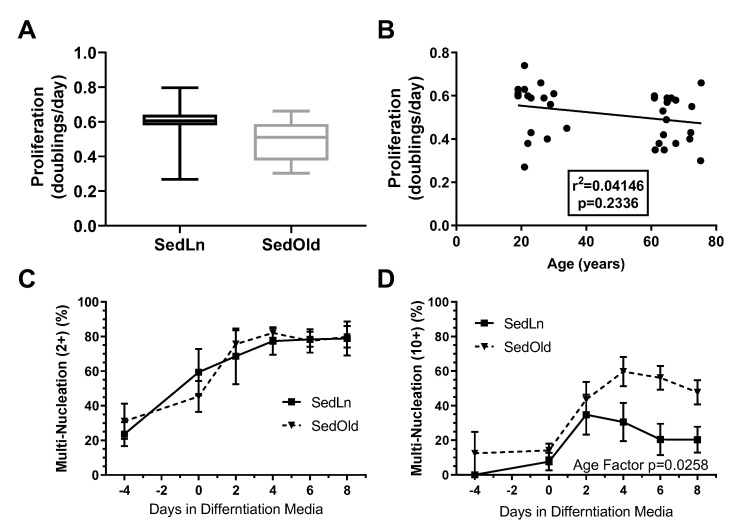
Proliferation rates (doubling/day) of satellite cells (SCs) isolated form sedentary lean (SedLn) (*n* = 18, age 19–34 years) and sedentary older (SedOld) (*n* = 20, age 61–76 years) participants (**A**). Correlation of SC proliferation rate and donor age (**B**). Fusion index scores ((**C**), 2+ nuclei per fiber, (**D**), 10+ nuclei per fiber) of SC from SedLn and SedOld donors. (* would indicate a *p*-value of <0.05).

**Figure 3 nutrients-12-02215-f003:**
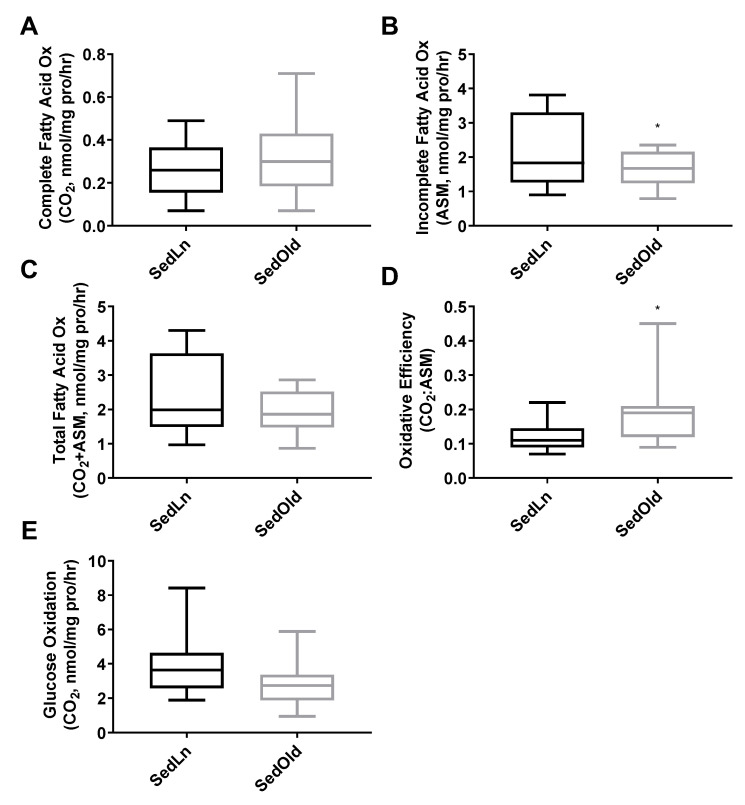
Complete (**A**), incomplete (**B**), and total (**C**) fatty-acid oxidation rates, as well as oxidative efficiency (complete/incomplete) (**D**), and glucose oxidation rates (**E**) of differentiated satellite cells from sedentary young (SedLn) and older donors (SedOld) (*n* = 5, BMI 18–25 kg/m+2+). (*, ** indicating a *p*-value of <0.05, <0.01, respectively).

**Figure 4 nutrients-12-02215-f004:**
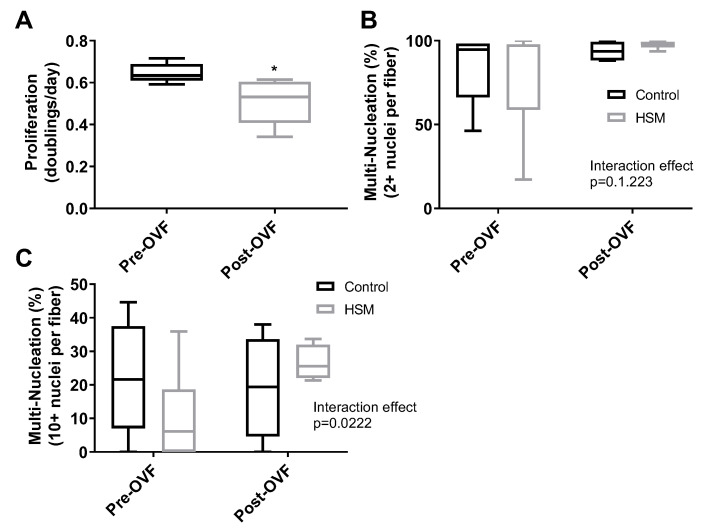
Proliferation rates (doubling/day) of satellite cells (SCs) isolated from human participants (*n* = 4) pre and post one week of overfeeding (OVF, 55% fat + 100 kcal) (**A**). Fusion index scores after eight days of differentiation ((**B**), 2+ nuclei per fiber, (**C**), 10+ nuclei per fiber) of SC from donors collected pre-OVF and post-OVF in control or high subsrate media (HSM) (+400 mM, 2:1, oleate–palmitate, +20 mM glucose). (*, ** indicating a *p*-value of <0.05, <0.01, respectively).

**Figure 5 nutrients-12-02215-f005:**
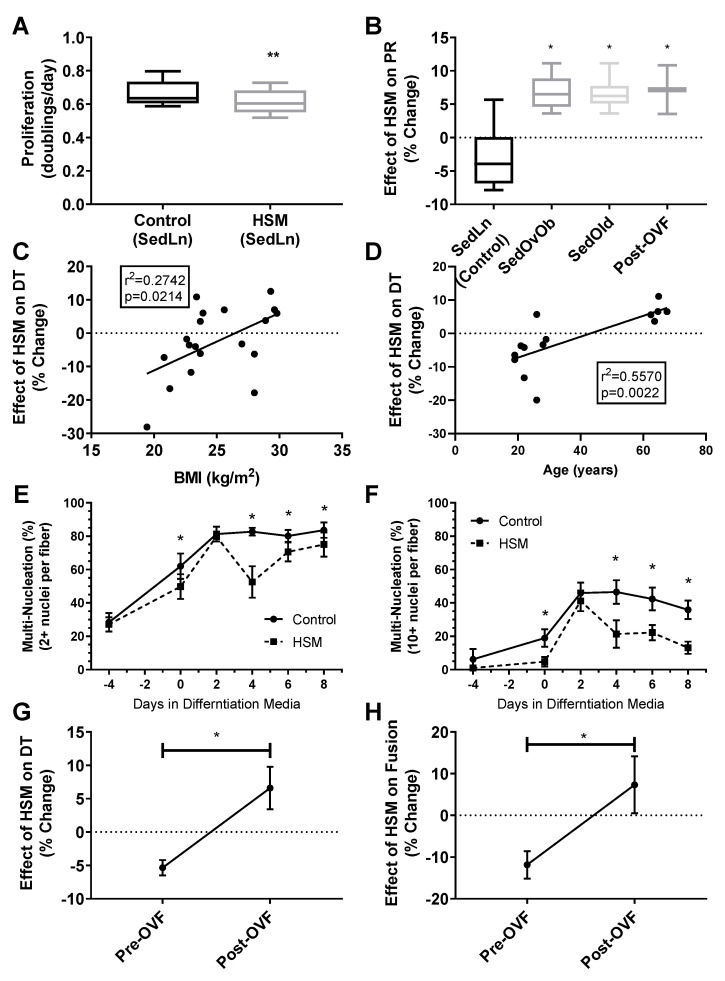
Proliferation rates (doubling/day) of satellite cells (SCs) isolated form sedentary lean (SedLn) (*n* = 10, BMI 18–25 kg/m^2^) participants treated with control vs. high substrate media (high substrate media (HSM), +400 mM, 2:1, oleate–palmitate, +20 mM glucose) (**A**). Response of SedLn (*n* = 10), sedentary overweight/obese (SedOvOb) (*n* = 5), post overfeeding (post-OVF) (*n* = 4) and sedentary old (SedOld) (*n* = 5). Effect (% change) of HSM media on SC proliferation rate (PR) of SedLn (*n* = 10), SedOvOb (*n* = 5), post-OVF (*n* = 4) and SedOld (*n* = 5) (**B**) groups, and correlated to SC donor BMI (SedLn, C) and age (SedLn and SedOld, (**D**)). Fusion index scores (**C**), 2+ nuclei per fiber (**E**), 10+ nuclei per fiber (**F**) of SC from SedLn donors treated with control or HSM media. Effect (% change) of HSM on proliferation rate (**G**) and eighth-day fusion index scores (**H**) on SC collected pre-OVF/post-OVF (*n* = 4/4). (*, ** indicating a *p*-value of <0.05, <0.01, respectively).

**Figure 6 nutrients-12-02215-f006:**
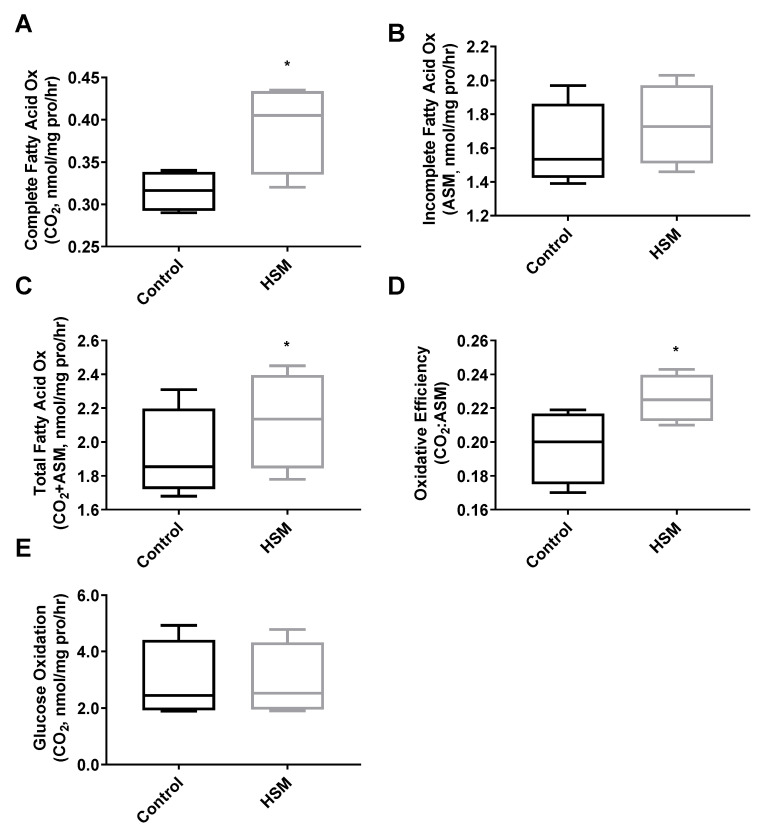
Complete (**A**), incomplete (**B**), and total (**C**) fatty-acid oxidation rates, as well as oxidative efficiency (complete/incomplete) (**D**) and glucose oxidation rates (**E**) of satellite cells (SC) from sedentary lean donors (*n* = 6, BMI 18–25 kg/m+2+) exposed to 4 days of control (growth media) or high substrate media (HSM) (growth media +400 mM, 2:1, oleate–palmitate, +20 mM glucose) prior to 7 days of serum-starve (2% horse serum) differentiation. (*, ** indicating a *p*-value of <0.05, <0.01, respectively).

**Figure 7 nutrients-12-02215-f007:**
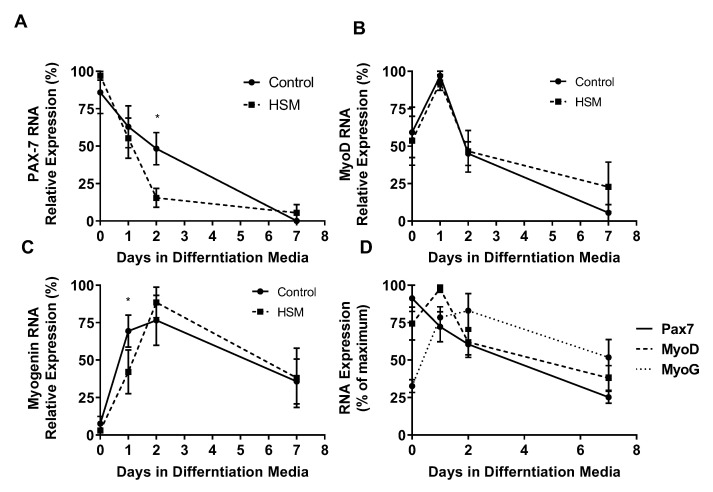
Effect of exposure to control or high substrate media (HSM, +400 mM, 2:1, oleate–palmitate, +20 mM glucose) on sedentary lean (SedLn) satellite cells (SCs) (*n* = 6) expression of myogenic factors; PAX-7 (**A**), MyoD (**B**), and MyoG (**C**), after 0, 1, 2, and 7 days of differentiation. Overlaid expression of PAX-7, MyoD, and MyoG throughout differentiation of control treated SC from SedLn donors (**D**). (*, ** indicating a *p*-value of <0.05, <0.01, respectively).

**Table 1 nutrients-12-02215-t001:** Anthropometric measures and blood metabolic markers. Bodyfat acquired through dual energy X-ray absorptiometry (DEXA) analysis. Blood metabolic markers were obtained from plasma isolated from participants after eight hours of fasting immediately prior to satellite cell isolations.

	SedLn	SedOvOb	SedOld	SedLn vs. SedOvOb	SedLn vs. SedOld
	*n* = 18	*n* = 5	*n* = 20		
Age (years)	23.6 ± 1	24.8 ± 1.9	66.3 ± 0.9	0.818	<0.001 ***
BMI (kg/m2)	23 ± 0.3	29.7 ± 1.8	26.5 ± 0.6	<0.001 ***	<0.001 ***
Bodyfat %	19.1 ± 1.5	32.8 ± 2.3	28.1 ± 1.3	<0.001 ***	<0.001 ***
Plasma Glucose (nM)	4.5 ± 0.3	5.1 ± 0.1	5.7 ± 0.3	0.634	0.022 *
Cholesterol (mg/dl)	147.3 ± 7.6	175 ± 10.3	195.8 ± 9.7	0.331	0.001 ***
HDL (mg/dL)	56.8 ± 3.1	45.3 ± 3.9	54.8 ± 2.9	0.186	0.849
LDL (mg/dL)	85.2 ± 6.1	120 ± 8.3	116.6 ± 8.8	0.125	0.014 *
Triglycerides (mg/dL)	90.6 ± 5.9	119.3 ± 12.7	121.8 ± 11.1	0.349	0.045 *

*, *p* < 0.05, ***, *p* < 0.001.

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
