# Peer review of "Overfeeding and Substrate Availability, But Not Age or BMI, Alter Human Satellite Cell Function"

_nutrients, 2020, doi:10.3390/nu12082215_

Round 1

Reviewer 1 Report

Fausnacht et al., investigate in vitro proliferation and differentiation rates of satellite cells obtained from young and old donors exposed to different dietary conditions and culture conditions. Overall, the results are of interest, but a lack of attention to detail and questionable group selections detract from the results. Figure order is mixed up and cytokine data is completely missing. Below are a list of comments and concerns.

High substrate media – How do the concentrations of glucose and fatty acids relate to what may be experienced in vivo? 25mM glucose would only be experienced in extreme cases.

‘Simple fusion’ is first mentioned on line 174, but not defined until line 192.

‘Robust fusion’ used on 176, and not defined.

Figure 1D is not referred to in the text.

Line 173-174: ‘SedLn did exhibit higher rates of simple fusion overall (p=0.04)’. This statistic is not represented in the figure and it is not clear what comparison is referred to. A change in rate would imply a significant change in delta (e.g. doublings/day) but Figure 1C displays the percentage of multi nucleation at each time point.   

It is unclear why the authors chose to count myotubes with 2+ nuclei and 10+ nuclei. In this case, 2+ nuclei also includes 10+ nuclei. Therefore, simple differentiation is misleading as it includes all differentiation. Typically, a fusion index would be used to describe such data with the proportion of cells in each differentiation phase reported (e.g. 2-3, 4-9, 10+).

Is there a difference between doublings per day (1A,B) and doublings per 24h (1E)? Again in figure 2.

Group selection for SedLn and SedOvOb is troubling. Firstly, there is not enough subjects that fit into the SedOvOb group (n=5) compared to n=20 for the SedLn group and there are really only 3 that show a markedly higher BMI than the rest of the subjects. Furthermore, in my opinion, BMI is a crude measure of obesity. Body fat % is a more pure measure of how ‘lean’ someone is. It is also odd that you see a correlation between proliferation rates and body-fat % but never separate subjects into low and high body fat %, despite there being a much more even spread of subjects with low and high body fat %.

Line 169-170. 23 young participants… but in Figure 1 you say n=20 SedLn and 5 SedOvOb…

Line 183-184. 23 young and 23 old subjects is 46… not 43.

Line 188-189: How were young and old cells matched based on body fat? It is completely unclear what has been done here.

Line 183-193. It is unclear why comparisons are made between SedOld and SedLn… This is comparing young apples with old oranges.  

Line 199. Does use of ‘significant’ refer to magnitude or statistical significance? If the former, please refrain from using significant in this context. If the later, please report p values and what the comparison refers to.

For all box plots please also display individual data points.

Line 215-216: ‘A reduction in both proliferation and fusion prior to differentiation treatment indicates cells which attempted to spontaneously differentiate were inhibited or perished.’ This statement is quite speculative. What evidence do you have to support this conclusion? Cells may also have not initiated spontaneous differentiation. A slower proliferation rate would also delay spontaneous fusion events as cell-to-cell proximity is a primary driver of spontaneous fusion.

Figure 5B. What is post-HFHVD? Is this post-OVF?

Line 248-249: ‘Autocrine excretion of inflammatory cytokines IL-6 and TNF-α also varied throughout proliferation and differentiation’. This data is completely missing???

Line 269. ‘A high fat/hypercaloric feeding intervention in humans negatively affected SC function’. It only reduced proliferation rate. Please be specific.

Line 270. BMI did not… but your subject numbers with high BMI are very low… and also body fat % did affect SC proliferation rate.

Line 313-314. ‘Donor body fat percentage had no relationship with in-vitro measures of SC proliferation or differentiation’. In Figure 1E you specifically show that there is a correlation between body fat % and SC proliferation rate…

315-316. Reference is missing.

Author Response

Response to Reviewer 1 Comments

Fausnacht et al., investigate in vitro proliferation and differentiation rates of satellite cells obtained from young and old donors exposed to different dietary conditions and culture conditions. Overall, the results are of interest, but a lack of attention to detail and questionable group selections detract from the results. Figure order is mixed up and cytokine data is completely missing. Below are a list of comments and concerns.

Figure Order has been corrected, data on cytokine secretion has been removed to make the manuscript more concise.

High substrate media – How do the concentrations of glucose and fatty acids relate to what may be experienced in vivo? 25mM glucose would only be experienced in extreme cases.

The concentration of 25mM glucose have been used before in cultured myoblast.  Concentrations of 5mM and 25mM DMEM are regularly produce by manufacturers and 25mM was used to ensure uniformity and reproducibility.  Both the growth and differentiation media lack insulin and the primary glucose transporter in these cells is GLUT1 (km ~1-3 mM).  Elevated glucose was also only used during the proliferative stage at which point GLUT4 is not highly expressed.  By best estimates maximal glucose uptake would peak around 6-8 mM with increasing concentrations not affecting transport rate.  Individuals with chronically elevated glucose routinely report fasting glucose above 6-8mM.  The use of 25mM as opposed to a smaller concentration ensures the cellular glucose supply remains constant with less frequent media changes as to not disrupt autocrine and paracrine secretion of growth factors.

‘Simple fusion’ is first mentioned on line 174, but not defined until line 192.

Clarified in methods section line 132

‘Robust fusion’ used on 176, and not defined.

Clarified in methods section line 133

Figure 1D is not referred to in the text.

Figure 1D has been referenced in the results

Line 173-174: ‘SedLn did exhibit higher rates of simple fusion overall (p=0.04)’. This statistic is not represented in the figure and it is not clear what comparison is referred to. A change in rate would imply a significant change in delta (e.g. doublings/day) but Figure 1C displays the percentage of multi nucleation at each time point.  

This analysis was generated as a treatment effect using one-way repeated measures ANOVA and has been clarified in the text and figure.

It is unclear why the authors chose to count myotubes with 2+ nuclei and 10+ nuclei. In this case, 2+ nuclei also includes 10+ nuclei. Therefore, simple differentiation is misleading as it includes all differentiation. Typically, a fusion index would be used to describe such data with the proportion of cells in each differentiation phase reported (e.g. 2-3, 4-9, 10+).

Due to fusion index not being standardized 2+ and 10+ were used to be most comparable with past and current studies.  2+ was used to detect any form of fusion which would indicate that the cells have undergone a phenotypic change and are no longer SCs while 10+ would more indicate myotube formation.  2+ was used instead of 2-9 because 2+ has been commonly used in older studies comparable to this paper.  It is agreed that all 10+ nuclei are included in the 2+ count.  Due to the quick turn around of this review it is not possible do a histogram representation of fusion as suggested but if the reviewer would like we can remove the 10+ data all together. 

Is there a difference between doublings per day (1A,B) and doublings per 24h (1E)? Again in figure 2.

Figure axis titles have been fixed to uniformly state doublings per day

Group selection for SedLn and SedOvOb is troubling. Firstly, there is not enough subjects that fit into the SedOvOb group (n=5) compared to n=20 for the SedLn group and there are really only 3 that show a markedly higher BMI than the rest of the subjects. Furthermore, in my opinion, BMI is a crude measure of obesity. Body fat % is a more pure measure of how ‘lean’ someone is. It is also odd that you see a correlation between proliferation rates and body-fat % but never separate subjects into low and high body fat %, despite there being a much more even spread of subjects with low and high body fat %. 

BMI was used because there are clearly defined categories for “Normal”, “Overweight”, and “Obese”, which do not exist with body fat %.  There is a limited number of overweight/obese samples which is why the correlation graphs for BMI and body fat were provided.

Line 169-170. 23 young participants… but in Figure 1 you say n=20 SedLn and 5 SedOvOb…

After reviewer suggestions to include blood measurements for all samples n sizes were reduced to only include samples for which all measurements were available.  Sample sizes have been fixed in text and in figures.

Line 183-184. 23 young and 23 old subjects is 46… not 43.

Fixed as above.

Line 188-189: How were young and old cells matched based on body fat? It is completely unclear what has been done here.

The measures of body fat matching have been excluded and returned to SedLn vs SedOld without bodyfat matching to simplify results.

Line 183-193. It is unclear why comparisons are made between SedOld and SedLn… This is comparing young apples with old oranges. 

SedLn and SedOld are matched based on activity level.  In many prior studies of young vs aged SCs activity level is not controlled.  It is thought that the novelty of this study is that it compares SC from sedentary young and sedentary old donors to make for a better comparison than pervious works which concluded stark differences in young active vs aged sedentary SCs.

Line 199. Does use of ‘significant’ refer to magnitude or statistical significance? If the former, please refrain from using significant in this context. If the later, please report p values and what the comparison refers to.

In the copy received line 199 refers to the legend for figure 2, but the word significant is meant to infer statistically significant and p-values have been added for each usage or the word changed.

For all box plots please also display individual data points.

This is not a standard practice of this journal but can be done if requested by the editor.

Line 215-216: ‘A reduction in both proliferation and fusion prior to differentiation treatment indicates cells which attempted to spontaneously differentiate were inhibited or perished.’ This statement is quite speculative. What evidence do you have to support this conclusion? Cells may also have not initiated spontaneous differentiation. A slower proliferation rate would also delay spontaneous fusion events as cell-to-cell proximity is a primary driver of spontaneous fusion.

This statement was not meant to be speculative but highlight the possibility that cells may have not survived during differentiation, artificially reducing proliferation measures. This has now been clarified.  Proliferation rate is often inversely correlated to fusion attempts as increased autocrine secretion of proliferative ligands (IL-6, EGF, IGF) suppress differentiation.  Many studies have shown that without close proximity fusion will fail but there is little mechanistic evidence for cell-to-cell proximity driving fusion and it is possible for cultured myoblast to reach near 100% confluency with little fusion. 

Figure 5B. What is post-HFHVD? Is this post-OVF?

Has been fixed to post-OVF

Line 248-249: ‘Autocrine excretion of inflammatory cytokines IL-6 and TNF-α also varied throughout proliferation and differentiation’. This data is completely missing???

Statement has been removed with the rest of cytokine data to make the results more concise.

Line 269. ‘A high fat/hypercaloric feeding intervention in humans negatively affected SC function’. It only reduced proliferation rate. Please be specific.

Clarified in text as recommended to only state effect on proliferation.

Line 270. BMI did not… but your subject numbers with high BMI are very low… and also body fat % did affect SC proliferation rate.

The limited sample size has been mentioned in the discussion text.  The emphasis on the null effect of BMI has been reduced throughout the text.  The influence of body-fat and BMI on proliferation and differentiation, respectively has been emphasized.

Line 313-314. ‘Donor body fat percentage had no relationship with in-vitro measures of SC proliferation or differentiation’. In Figure 1E you specifically show that there is a correlation between body fat % and SC proliferation rate…

This has been fixed in the text as recommended.  The correlation of body fat and PR has been addressed in the results and discussion.

315-316. Reference is missing.

Reference has been added as recommended.  

Reviewer 2 Report

  1. Line 86: The sentence ‘sedentary individuals with overweight/obese’; did you mean ‚sedentary and overweight or obese individuals’?
  2. Line 81/82: Skeletal muscle biopsies were obtained under IRB-approved protocols (#14-1234 and #11-077). Please define the abbreviation and describe the ethical approval according to journals’ requirements: When reporting on research that involves human subjects, human material, human tissues, or human data, authors must declare that the investigations were carried out following the rules of the Declaration of Helsinki of 1975, revised in 2013. At a minimum, a statement including the project identification code, date of approval, and name of the ethics committee or institutional review board should be cited in the Methods Section of the article. Did the participants give written informed consent? Please describe.
  3. Please provide BMI range for all groups.
  4. Line 90: Please provide baseline characteristics of SC donors, i.e. weight, fasting blood glucose, cholesterol and triglyceride levels etc.
  5. How did dietary intervention change body weight and other metabolic parameters? Is there any correlation with in vitro results?
  6. Please pay attention to uniform formatting of the references in the text.
  7. Line 136: Shouldn’t it read ‘the effects of’?
  8. Line 144: Shouldn’t it read ‘as stated’?
  9. Line 145: Shouldn’t it mean ‘was collected 2 days before’?
  10. Line 145/146: Please delete the doubling ‘For each collection’.
  11. Please pay attention to the numbering and location of figures (e.g. Fig.6 is located between Fig.4 and Fig.5; Fig.3 is located after Fig. 7).
  12. Please include the number of samples/participants for each figure in the figure legends.
  13. 5C and D: Why don’t the authors show the data of all samples/participants? In Fig. 5C there are 19 dots, in Fig. 5D 14, while in the figure legend of Fig.5 n=20 is stated. Please explain.
  14. Line 343: Please delete ‘did’.
  15. 7A-D: Shouldn’t it read ‘Days in Differentiation media’.
  16. Could you provide a (literature) comparison with data from trained subjects?
  17. Can the authors discuss data (own or literature) or just speculate about changes of results if a high-carbohydrate diet instead of high-fat diet would have been used prior to SC sampling?

Author Response

Response to Reviewer 2 Comments

Line 86: The sentence ‘sedentary individuals with overweight/obese’; did you mean ‚sedentary and overweight or obese individuals’?

Sentence has been clarified as recommended.  There are three groups all sedentary 1) young and lean 2) young and overweight or obese, and 3) older and lean.

Line 81/82: Skeletal muscle biopsies were obtained under IRB-approved protocols (#14-1234 and #11-077). Please define the abbreviation and describe the ethical approval according to journals’ requirements: When reporting on research that involves human subjects, human material, human tissues, or human data, authors must declare that the investigations were carried out following the rules of the Declaration of Helsinki of 1975, revised in 2013. At a minimum, a statement including the project identification code, date of approval, and name of the ethics committee or institutional review board should be cited in the Methods Section of the article.

Studies were approved via the Virginia Polytechnic Institute and State University Internal review board.  The statement has been clarified and protocol numbers were given in the text

Did the participants give written informed consent? Please describe.

Written consent was obtained copies of the written consent forms have been given to the editor

Please provide BMI range for all groups.

BMIs have been added as requested.

Line 90: Please provide baseline characteristics of SC donors, i.e. weight, fasting blood glucose, cholesterol and triglyceride levels etc.

A table has been added with requested measures off blood glucose TAG, HDL, LDL and cholesterol.

How did dietary intervention change body weight and other metabolic parameters? Is there any correlation with in vitro results?

The dietary intervention caused an increase in weight gain and body fat which has been added to the results (line213-217).

Please pay attention to uniform formatting of the references in the text.

References have been universally formatted as recommended.

Line 136: Shouldn’t it read ‘the effects of’?

Changed as recommended.

Line 144: Shouldn’t it read ‘as stated’?

Changed as recommended.

Line 145: Shouldn’t it mean ‘was collected 2 days before’?

Section has been removed

Line 145/146: Please delete the doubling ‘For each collection’.

Section has been removed

Please pay attention to the numbering and location of figures (e.g. Fig.6 is located between

The figure placement was changed by the journal editor based on size.  Numbering has been reordered to match editor’s placement.

Fig.4 and Fig.5; Fig.3 is located after Fig. 7).

The figure placement was changed by the journal editor based on size.  Numbering has been reordered to match editor’s placement.

Please include the number of samples/participants for each figure in the figure legends.

The n-sizes have been added for all figures.

5C and D: Why don’t the authors show the data of all samples/participants? In Fig. 5C there are 19 dots, in Fig. 5D 14, while in the figure legend of Fig.5 n=20 is stated. Please explain.

After reviewer suggestions to include blood measurements for all samples n sizes were reduced to only include samples for which all measurements were available.  Sample sizes have been fixed in text and in figures.  In some instance data points are overlapping on graphs.

Line 343: Please delete ‘did’.

Deleted as recommended

7A-D: Shouldn’t it read ‘Days in Differentiation media’.

Changed as recommended

Could you provide a (literature) comparison with data from trained subjects?

A second manuscript is in production that highlights the differences between sedentary and trained subjects but is not yet available.  This data shows significant differences in the proliferation rates and fusion patterns of SCs from highly trained distance runners.  Sedentary vs trained SCs show distinctive differences in cytokine secretion during proliferation and differentiation and a profoundly and trained SCs have a profoundly negative response to HSM.

Can the authors discuss data (own or literature) or just speculate about changes of results if a high-carbohydrate diet instead of high-fat diet would have been used prior to SC sampling?

This section has been added as requested.  Many prior groups have shown that manipulating beta oxidation has a controlling effect on SC differentiation and performance.  The authors of this study speculate that because high carb diets have a less pronounced effect on oxidation, they would not reduce proliferation and differentiation to the same degree.  Most proliferating stem cells, including SCs, are mostly glycolytic and even anaerobic.  Conditioning cells with a high carb diet prior to isolation may increase glucose uptake which could enhance proliferation and delay spontaneous fusion.  Most animal models including mouse and rat but especially pig are regularly fed a relatively high carbohydrate but yet yield highly functioning SC when cultured. 

Round 2

Reviewer 1 Report

I thank the authors for addressing my concerns and comments. I now find the manuscript suitable for publication.